# Analysis of Intrinsic Mechanistic of Stability-Tracking Control for Distributed Drive Autonomous Electric Vehicle

Xuequan Tang [1], Yunbing Yan [1,*], Baohua Wang [2], Xiaowei Xu [1] and Lin Zhang [1]

1 Department of Automobile and Traffic Engineering, Wuhan University of Science and Technology, Wuhan 430065, China; tang.x.q@foxmail.com (X.T.); xuxiaowei@wust.edu.cn (X.X.); zhanglin4025@wust.edu.cn (L.Z.)
2 School of Automotive Engineering, Hubei University of Automotive Technology, Shiyan 442002, China; 19950009@huat.edu.cn
* Correspondence: yanyunbing@wust.edu.cn

**Abstract:** For distributed drive autonomous vehicles, adding lateral stability control (LSC) to the trajectory tracking control (TTC) can optimize the distribution of the driving torque of each wheel, so that the vehicle can track the planned trajectory while maintaining stable lateral motion. However, the influence of adding LSC on the TTC system is still unclear. Firstly, a stability-track hierarchical control structure composed of LSC and TTC was established, and the interaction between the two layers was identified as the key of this paper. Then, the Intrinsic Mechanistic framework of the stability-tracking control (STC) was proposed by establishing and analyzing the vehicle dynamic model and control process of two layers. Finally, through simulation experiments, it was found that the change in the curvature of the target trajectory will make the tracking target trajectory and maintaining the lateral stability of the vehicle appear to conflict; in addition, in the LSC layer, the steering characteristics and delay characteristics of different reference models have a greater impact on the lateral stability and trajectory tracking performance; moreover, adjusting the preview time has a more obvious effect on trajectory tracking and lateral stability than the stability correction intensity coefficient.

**Keywords:** distributed drive autonomous vehicles (DDAVs); lateral stability; vehicle lateral stability control (LSC); trajectory tracking control (TTC)

## 1. Introduction

Autonomous driving vehicles (ADVs) is an advanced vehicle intelligent technology that integrates environment perception, decision planning and motion control in virtue of intelligent equipment [1]. Due to the simple transmission mode of pure electric vehicles, most of the ADVs use pure electric chassis. In many driving modes of electric vehicles, the distributed driving mode with hub motor drive as the core has unique advantages in enhancing vehicle safety, handling stability and improving vehicle energy efficiency, and has become a hot spot in academic and industrial circles [2,3]. By optimizing the allocation of the driving/braking torque of each wheel, many vehicle active safety technologies can be implemented through this structure, such as Antilock Braking System (ABS), Acceleration Slip Regulation (ASR), Differential Drive Assist Steering (DDAS), Electronic Differential Control (EDC), Four Wheel Steering (4WS), Vehicle Stability Control (VSC), etc., [4–6]. Direct Yaw-Moment Control (DYC) is the most effective and common control method for LSC [7,8]. The main principle is to generate different longitudinal forces in each wheel of the vehicle to produce inward or outward yaw moment to maintain stability of the vehicle [9]. Distributed drive electric vehicles (DDEVs) simplify the vehicle transmission system and have the advantage that the motor can be controlled independently, quickly and precisely, and DYC can be easily achieved by allocating driving/braking torque to each wheel [8,10].

Trajectory tracking control (TTC) is one of the key technologies of ADVs control system. The objective of TTC is to achieve the vehicle following the planned path with small lateral displacement error and other performance requirements through the vehicle's steering and drive control subsystem. A large number of trajectory tracking algorithms have been applied to autonomous motion control subsystems, such as Stanley Model, Optimal Preview Control (OCM), Proportion Integration Differentiation (PID), Model predictive control (MPC), linear quadratic regulator (LQR), Sliding Mode Control (SMC), H∞ control, Neural Network Model (NNM), etc., [11,12]. However, these algorithms only provide the vehicle with the ability to perform direction and speed control under good road adhesion conditions, but they seldom consider the lateral stability under extreme conditions. When ADVs vehicle travels on a road with poor adhesion conditions and high vehicle speed, the vehicle may become unstable and drive out and into the target trajectory [13,14].

In some of the literature, LSC has been added to the TTC algorithm in an attempt to improve the vehicle's ability to maintain lateral stability for TTC in extreme path-following scenarios. Here, we summarize the control system composed of TTC and LSC as stability tracking control (STC). Wu et al. [15] designed a trajectory tracking controller based on MPC, which can allocate the steer angle and the torque of each wheel, while considering the actuator constraints and the vehicle stability constraints, and then track the desired trajectory. The controller effectively realizes trajectory tracking with high accuracy and lateral stability and strong robustness. Similarly to Reference [15], Reference [13] adopted the method of multi-objective control, using sideslip angle and yaw rate as the reference input of stability control target and vehicle speed; and yaw angle and lateral displacement as maneuverability and tracking accuracy, respectively. The main innovation of this approach is to transform the constraint problem of the lateral motion state of the vehicle into the planning and tracking problem of the lateral motion state. In contrast to the above two studies, in the literature [14,16–22], a hierarchical control approach was used to decouple the speed tracking control and TTC, as well as the LSC of the vehicle. By comparing the simulation results of decoupling or not, it was found that the decoupling approach had better speed-tracking performance in the results [17]. In the literature [16,18,19,21], different speed control, LSC and TTC were proposed, using different control algorithms for the issues caused by the nonlinear, high coupling and over-actuated characters of DDEVs vehicles. Hu et al. [22] proposed an output constraint controller, which enables the vehicle to maintain STC without exceeding the safety boundary, using the robust LQR algorithm. Chen et al. [20] proposed a super-twisting second-order sliding mode control algorithm with the nonlinear disturbance observer when considering the lateral slope of the road. Liang et al. [14] pointed out the strong coupling between longitudinal motion and yaw motion, where the path-following objectives and dynamical stability may strongly conflict with each other, and they proposed a path following controller based on yaw rate prediction and a coordinated control mechanism to achieve the coordinated control of path-following and lateral stability. In a word, these studies confirm that TTC and LSC need to be coordinated. However, the intrinsic mechanism of STC is not clear. Research on this issue has the following significance: (1) by analyzing whether the TTC and LSC will conflict under their respective control objectives, guidance can be put forward for the stability-tracking coordination control method; (2) and by analyzing the process of STC, the main factors affecting STC can be found, and the efficiency of the STC system can be improved.

In order to obtain the principle of the interaction of the two control layers and the main factors affecting the STC, a framework of interaction mechanism between TTC and LSC is proposed. In the process of illustrating and discussing the contents of this framework, a vehicle dynamics model is developed and a STC method is proposed. In the TTC layer, the optimal preview control model is used to obtain the front wheel steer angle. In the LSC layer, the additional yaw moment is calculated by using five reference model, and a torque allocation method combining the proportional allocation rule based on axial load and

minimizing tire utilization rule is proposed to calculate the torque of each wheel. Finally, the effects of target trajectory, reference model and control parameters on the STC system are analyzed qualitatively and quantitatively through simulation experiments.

## 2. Stability-Tracking Hierarchical Control Structure

Hierarchical control structure is widely used in vehicle dynamics chassis integrated control and the intelligent driving control system. In the design of the intelligent driving control system, the whole system is often divided into perception layer, decision-making and planning layer, motion control layer, execution layer and so on [23,24]. At present, in DDEVs, LSC is mainly implemented by a hierarchical mechanism, which is generally divided into three layers: vehicle motion state prediction layer, vehicle motion tracking layer, and torque distribution and executive layer [2,3,25–27].

According to the reviewed references, the structure combining vehicle LSC and TTC of DDEVs is simplified into a stable-tracking hierarchical control structure [14,16–22], as shown in Figure 1. The control structure mainly includes a trajectory planning layer; trajectory tracking layer; motion tracking or stability discriminant layer; torque allocation layer; and execution layer composed of motors, tires and other subsystems of the vehicle. Here, the target trajectory given by the trajectory planning layer is simplified and represented by a trajectory line. The function of the trajectory tracking layer is to control the vehicle to track the target trajectory line and calculate the steer wheel angle and braking/acceleration force. The function of the LSC layer is to calculate the additional yaw moment and longitudinal force required by the vehicle according to the acceleration/braking force and steer angle. In this process, the feedback information of vehicle state will affect the judgment of both the two layers, and the steering wheel angle obtained by the trajectory tracking layer will also directly affect the LSC layer. Obviously, in this structure composed of multilayer feedforward and multilayer feedback, the interaction between the TTC layer and the LSC layer will affect the performances of trajectory tracking and vehicle stability. Therefore, the key to the study of stability-tracking control system is to study the interaction between TTC layer and vehicle LSC layer.

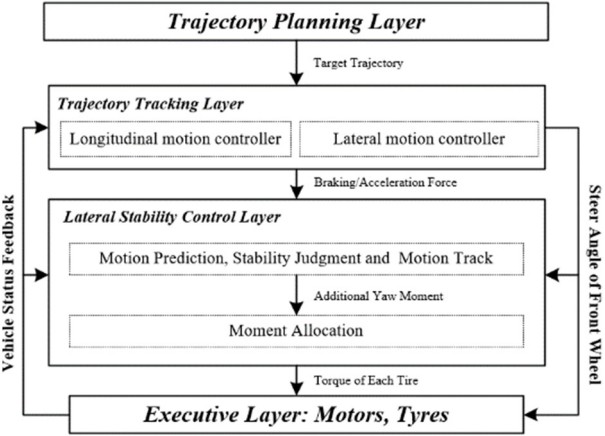

**Figure 1.** Stability-track hierarchical control structure.

## 3. Analysis of the Interaction Process between Trajectory Tracking Control and Lateral Stability Control

In order to analyze the interaction process between the TTC layer and the LSC layer, the vehicle dynamics model is used as a motion reference model to predict the motion state and trajectory of the vehicle. The TTC layer consists of a lateral displacement tracking controller and a longitudinal velocity tracking controller [11]. The LSC layer consists of prediction of the vehicle state, motion tracking and torque allocation. The research in this paper belongs to primal research. Consequently, the control model and the control structure and parameters should be representative and not complex.

### 3.1. Vehicle Dynamic Model

Due to its simplicity, the yaw plane vehicle model considering the lateral movement, longitudinal movement and heading angle change of the vehicle is often used in the literature [13,17,18,20]. The yaw plane dynamics vehicle model is established in this paper, as shown in Figure 2. The origin of the vehicle body coordinate system is the center of mass, the *X*-axis points to the heading direction and the *Y*-axis points to the left side in the top view. The vehicle sideslip angle, yaw rate and longitudinal velocity of the center mass were selected as motion state variables, and the front wheel steer angle and the driving/braking torque of the four tires were selected as the control input of the system. The symbols and physical quantities in the modeling process are listed in Table 1.

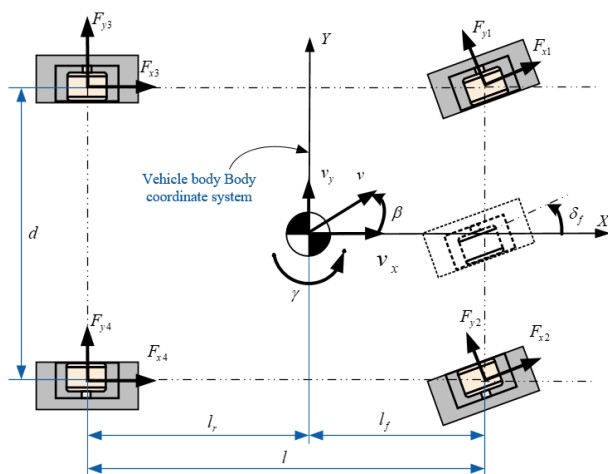

**Figure 2.** Yaw plane dynamics vehicle model.

**Table 1.** Symbols of vehicle.

| Symbol | Description | Value |
|---|---|---|
| $m$ | Vehicle mass (kg) | 1830 |
| $vx, vy$ | Longitudinal and lateral velocity (km/h) | - |
| $lf, lr$ | Distance from center mass to front and rear axle (m) | 1.4, 1.6 |
| $l$ | Distance from front axle to rear axle (m) | |
| $d$ | half of the vehicle front and rear track widths (m) | 1.6 |
| $hg$ | Height of center mass (m) | 0.5 |
| $kfl, kfr, krl, krr$ | Cornering stiffness of four tires (N/rad) | - |
| $\alpha fr, \alpha fr, \alpha rl, \alpha rr$ | Slip angle of four tires (rad) | - |
| $\beta$ | Sideslip angle of vehicle (rad) | - |
| $g$ | Acceleration of gravity (m/s²) | 9.8 |
| $\gamma$ | Yaw rate of the sprung mass of the tractor (rad/s) | - |
| $IZ$ | Yaw moment of inertia of center mass (kg·m²) | 3655.4 |
| $m$ | Vehicle mass (kg) | 1830 |

According to Newtonian mechanics, the yaw plane dynamics model of the vehicle can be established.

Longitudinal motion equation:

$$m(\dot{v}_x - \gamma \cdot v_y) = (F_{xfl} + F_{xfr})\cos\delta_f - (F_{yfl} + F_{yfr})\sin\delta_f + F_{xrl} + F_{xrr} \tag{1}$$

Lateral motion equation:

$$m(\dot{v}_y + \gamma \cdot v_x) = (F_{xfl} + F_{xfr})\sin\delta_f + (F_{yfl} + F_{yfr})\cos\delta_f + F_{yrl} + F_{yrr} \tag{2}$$

Yaw motion equation:

$$\begin{aligned} I_Z \cdot \dot{\gamma} \quad &= l_f(F_{xfl} + F_{xfr})\sin\delta_f + l_f(F_{yfl} + F_{yfr})\cos\delta_f - \\ &l_r \cdot (F_{yrl} + F_{yrr}) + \tfrac{d}{2}(F_{xfr} - F_{xfl})\cos\delta_f + \tfrac{d}{2}(F_{xrr} - F_{xrl}) \\ &+ \tfrac{d}{2}(F_{yfl} - F_{yfr})\sin\delta \end{aligned} \tag{3}$$

The cornering force of each tire is as follows:

$$F_{yi} = k_i \cdot \alpha_i \tag{4}$$

where $i = fl, fr, rl$, and $rr$ represents front left wheel, front right wheel, rear left wheel and rear right wheel, respectively. The sideslip angle of front and rear axle tires is calculated as follows:

$$\begin{cases} \alpha_{fl,fr} = \beta + \dfrac{l_f \gamma}{v_x} - \delta_f \\ \alpha_{rl,rr} = \beta - \dfrac{l_r \gamma}{v_x} \end{cases} \tag{5}$$

The tire lateral dynamics model is the most common way to describe the interaction between the road surface and the tire. Figure 3 shows the tire cornering force curve under different loads, using empirical data in CarSim. The slope of the curve is the tire cornering stiffness, which can be divided into linear and non-linear regions. Under the assumption of a small sideslip angle, the relationship between the cornering force and the sideslip angle of the tire under small lateral acceleration can be easily expressed with the linear tire model used [14].

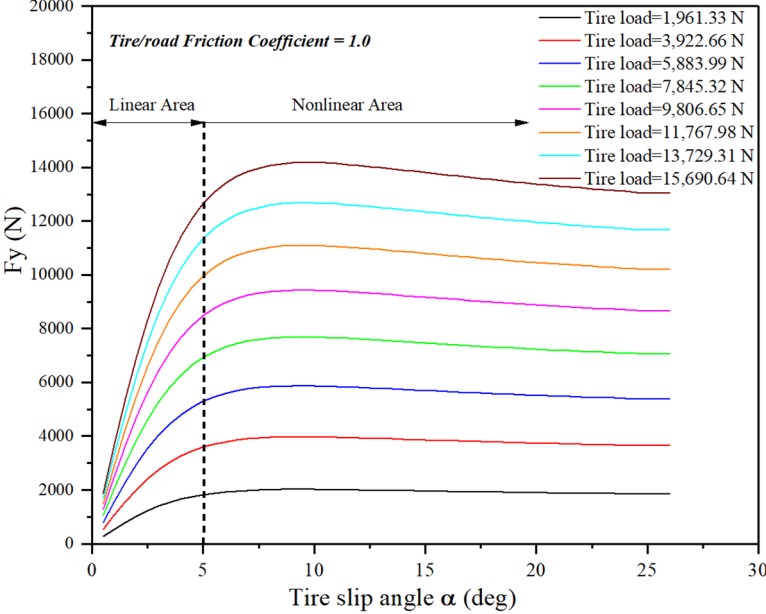

**Figure 3.** Tire cornering force curves under different loads.

A linear tire model under a specific tire load can be constructed by linear interpolation, and the formula for calculating tire cornering force is as follows:

$$F_y = F_y(j+1) \cdot \frac{F_{zi} - F_z(j)}{F_z(j+1) - F_z(j)} + F_y(j) \cdot \frac{F_z(j+1) - F_{zi}}{F_z(j+1) - F_z(j)} \tag{6}$$

The combined cornering stiffness of the front and rear axle tires is as follows:

$$k_i = \frac{\sum\limits_{k=1}^{n} F_{yk}/\alpha_k}{n} \tag{7}$$

where $F_z(j)$ ($j$ = 1, 2, 3, ...8) is the vertical load corresponding to curves, and $F_y(j)$ is the cornering force curve in Figure 3. $F_{zi}$ is the vertical load of each tire, as follows [8,17,19]:

$$\begin{cases} F_{Zfl} = \frac{m}{l}\left(\frac{1}{2}gl_r - \frac{1}{2}\dot{v}_x h_g - \frac{l_r}{d}\dot{v}_y h_g\right) \\ F_{Zfr} = \frac{m}{l}\left(\frac{1}{2}gl_r - \frac{1}{2}\dot{v}_x h_g + \frac{l_r}{d}\dot{v}_y h_g\right) \\ F_{Zrl} = \frac{m}{l}\left(\frac{1}{2}gl_f + \frac{1}{2}\dot{v}_x h_g - \frac{l_f}{d}\dot{v}_y h_g\right) \\ F_{Zrr} = \frac{m}{l}\left(\frac{1}{2}gl_f + \frac{1}{2}\dot{v}_x h_g + \frac{l_f}{d}\dot{v}_y h_g\right) \end{cases} \tag{8}$$

From (1) to (8), it can be seen that the front wheel steer angle and the driving/braking torque of each wheel produces the power of vehicle forward/braking, yaw motion and lateral motion. The change of vertical load will cause the change of cornering stiffness each tire, thus leading to the change of tire cornering force. The additional yaw moment reduces the ratio of the yaw moment generated by the tire cornering and makes the tire cornering force as close to the linear region as possible. These forces and torques will affect the longitudinal movement, lateral movement and yaw movement of the vehicle. From the perspective of vehicle dynamics, the three are coupled with each other. There are two reasons for coupling, namely input coupling and motion component coupling. The relationship between the product of the longitudinal force of each tire and the front wheel steer angle is the input coupling. The acceleration components of lateral and longitudinal motions (the second term on the left side of the Equations (1) and (2)) are the motion component coupling. The two reasons directly affect the change of vehicle motion state and the feedback of motion state.

### 3.2. Trajectory Tracking Control Layer

The TTC layer is composed of speed controller and direction controller. In this section, we establish the optimal preview control driver model, which is a remarkably representative steering control method [12,28]. Figure 4 is the schematic diagram of the optimal control driver model. Its coordinate system includes a global coordinate system, inertial coordinate system and vehicle coordinate system. The origin of the vehicle coordinate system is fixed at the center of the front axle, and the $X_V$ axes and $Y_V$ axes are parallel to the longitudinal axis and the first axle of the vehicle, respectively. The origin of the inertial coordinate system coincides with the origin of the vehicle coordinate system; the $X_i$ axis and $Y_i$ axis are parallel to the *X*-axis and *Y*-axis of the global coordinate system, respectively; and the included angle between the front of the vehicle and the $X_i$ axis of the inertial coordinate system is the yaw angle, $\psi$.

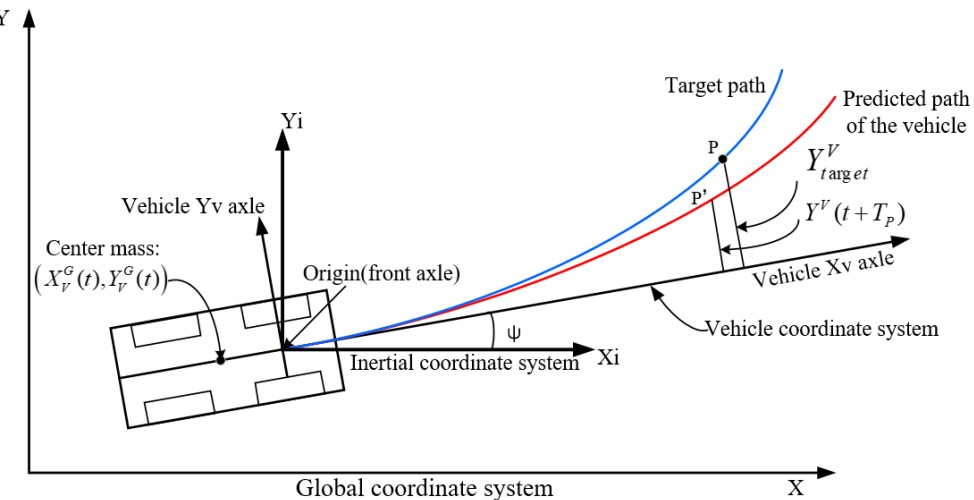

**Figure 4.** Schematic diagram of optimal preview control driver model.

Using the coordinate transformation method, we see that the preview point, P, in the vehicle coordinate system and the ordinate of the vehicle at $t + T_{PT}$ can be obtained as follows:

$$\begin{cases} Y_{target}^V = [Y_{target}^G - Y_V^G(t)] \cos \psi - [X_{target}^G - X_V^G(t)] \sin \psi \\ Y^V(t + T_{PT}) = [Y_V^G(t + T_{PT}) - Y_V^G(t)] \cos \psi - [X_V^G(t + T_{PT}) - X_V^G(t)] \sin \psi \end{cases} \tag{9}$$

where the superscript of the symbol represents the coordinate system it is in, and the subscript represents the object it belongs to. $X_{target}^G$ and $Y_{target}^G$ are respectively the abscissa and ordinate of the preview point, P. The abscissa of the preview point in the vehicle coordinate system is the product of preview time, $T_{PT}$, and vehicle speed, $v_x$.

The modeling process of the Optimal Preview Control (OPC) driver model is explained in detail in Reference [29], and the general modeling process is as follows:

Firstly, assuming that the speed is constant, a 2-DOF linear dynamics model of vehicle yaw motion is established to predict vehicle motion state, and its state space expression is as follows:

$$\begin{bmatrix} \dot{Y}_V^G \\ \gamma \\ a_y \\ \dot{\gamma} \end{bmatrix} = \begin{bmatrix} 0 & v_x & 1 & 0 \\ 0 & 0 & 0 & 1 \\ 0 & 0 & \frac{-(k_f+k_r)}{mv_x} & \frac{l_r k_r - l_f k_f}{mv_x} - v_x \\ 0 & 0 & \frac{l_r k_r - l_f k_f}{I_z v_x} & \frac{-(l_f^2 k_f + l_r^2 k_r)}{I_z v_x} \end{bmatrix} \begin{bmatrix} Y_V^G \\ \psi \\ v_y \\ \gamma \end{bmatrix} + \begin{bmatrix} 0 \\ 0 \\ \frac{k_f}{m} \\ \frac{l_f k_f}{I_z} \end{bmatrix} [\delta_f] \tag{10}$$

where $k_f$ and $k_r$ are the total cornering stiffness of tires of the front and rear axles, respectively, and $X(t) = \begin{bmatrix} Y_V^G(t) & \psi & v_y & \gamma \end{bmatrix}^T$ is the motion status of the vehicle at time, $t$. The lateral displacement of the vehicle front axle center is as follows:

$$Y_f^G(t) = \begin{bmatrix} 1 & a & 0 & 0 \end{bmatrix} \cdot \begin{bmatrix} Y_V^G(t) & \psi & v_y & \gamma \end{bmatrix}^T \tag{11}$$

Then, the lateral displacement of the vehicle front axle center at time, $t + T_{PT}$, is predicted as follows:

$$Y_f^G(t + T_{PT}) = Ce^{A \cdot T_{PT}} \cdot X(t) + C \int_0^{T_{PT}} e^{A\eta} d\eta Bu \tag{12}$$

where $A \in R4 \times 4$, $B \in R4 \times 1$ and $C \in R1 \times 4$ are the state matrix, control matrix and coefficient matrix of output vector of the state space expression (10), respectively. The first and second terms on the right side of the equation correspond to the free response and forced response of the vehicle motion, respectively.

Finally, the objective function of minimizing the sum of squares of track tracking errors was established to solve the optimal wheel steer angle:

$$J = \frac{1}{T_{PT}} \int_t^{t+T_{PT}} \left( Y_{target}^V(\eta) - Y^V(t + T_{PT}) \right)^2 w(\eta) d\eta \tag{13}$$

The optimal wheel steer angle after discretization is as follows:

$$\delta_f(t + t_d)^* = \delta_f(t) + \frac{Y_{target}^V - F * X(t)}{G} \tag{14}$$

where we have the following:

$$\begin{cases} G = C \int_0^{T_{PT}} e^{A\eta} d\eta B \\ F = C e^{A \cdot T_{PT}} \end{cases} \tag{15}$$

In Formula (14), the denominator of the second term on the right of the equation is the difference between the lateral displacement deviation of the preview point, P, and the displacement of the free response of the vehicle, and the numerator $G \in R1 \times 1$ is the coefficient of the forced response.

Obviously, the key to maintaining stable steering lies in the magnitude of $G$, which is a function of TP. The relationship between the preview time ($T_{PT}$) and $G$ at different vehicle speeds ($v_x$) can be obtained by bringing in the necessary parameters, as shown in Figure 5. It can be seen from the figure that $G$ gradually increases with the increase of $T_{PT}$ and $v_x$, and finally remains unchanged. According to Formula (14), a smaller $G$ value results in a larger increment of front wheel steer angle and a larger rate of change.

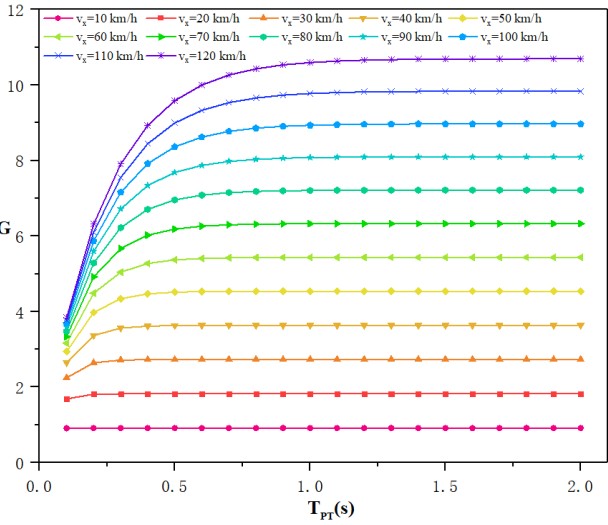

**Figure 5.** The relationship between the preview time $T_{PT}$ and $G$.

A big change rate of front wheel steer angle will cause the vehicle's motion state to be close to the instability limit. Figure 6 shows the phase plane portrait of yaw rate and sideslip angle of the vehicle at different rate of the front wheel steer angle. When the total wheel angle is constant, the larger the rate, the larger the area of the phase curve enclosed by yaw rate and sideslip angle of the vehicle, and the easier the vehicle state is to approach the instability boundary in the process of transferring from stable point P1 to another stable point, P2.

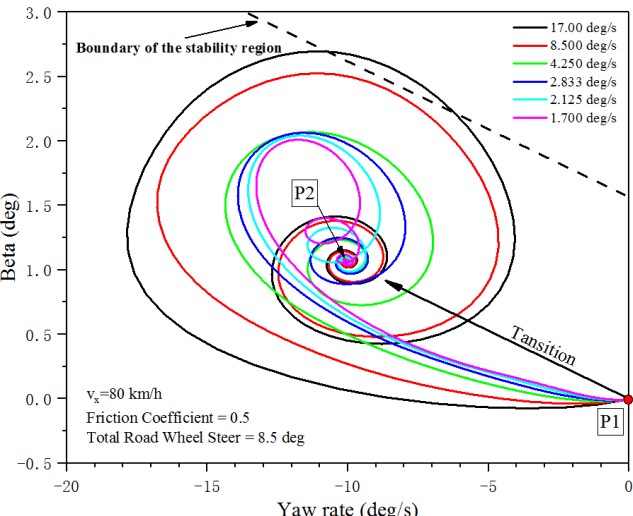

**Figure 6.** Phase plane portrait of yaw rate and sideslip angle of the vehicle at different rate of the front wheel steer angle.

Based on the above analysis, if the parameter $T_{PT}$ of the TTC is too small, it may make the stability of the vehicle worse.

The longitudinal speed tracking controller is designed by using the PID algorithm, and the driving/braking force is then calculated. The constant speed, vxr, is defined as the target speed; and the difference between the current speed, $v_x$, and the target speed, $v_{xr}$, is as follows:

$$\Delta v_x = v_{xr} - v_x \tag{16}$$

The driving/braking force is as follows:

$$F_x = K_T[\Delta v_x + \frac{1}{T_{iT}}\int_0^t \Delta v_x dt + T_{dT}\frac{d\Delta v_x}{dt}] \tag{17}$$

where $K_T$, $T_{iT}$ and $T_{dT}$ are proportional coefficient, integral time and differential time, respectively.

The speed and steering control process of the trajectory tracking controller can be simplified into a structure composed of a prediction layer and tracking layer, as shown in Figure 7.

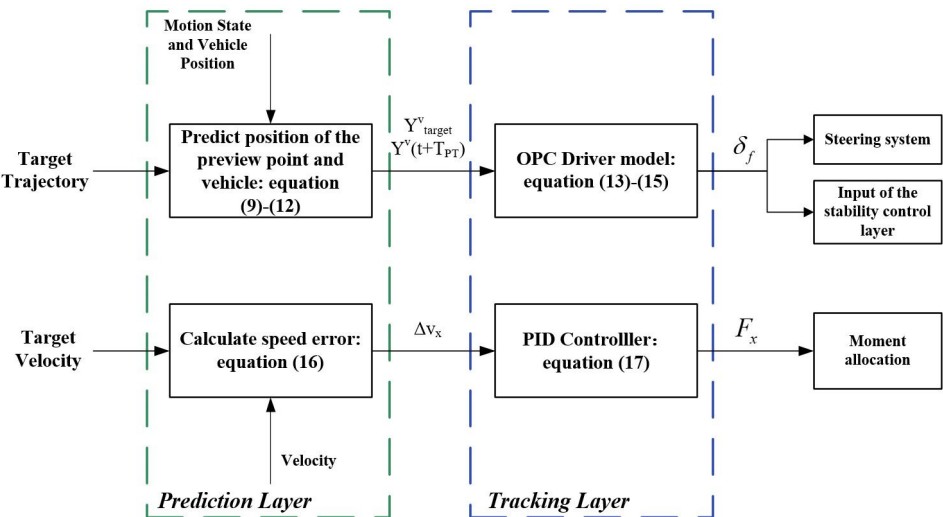

**Figure 7.** Structure of trajectory tracking control layer.

The vehicle state at the current moment and the target trajectory jointly determine the predicted results in the future period of time.

### 3.3. Lateral Stability Control Layer

In most of the current literature, LSC uses a control logic based on certain conditional activation; that is, Direct Yaw Control is applied when the yaw rate of the vehicle exceeds a certain threshold value [9]. It is difficult to continuously study the influence of yaw moment on trajectory tracking performance in this way. In this paper, an unconditional normally open mode is adopted; that is, whether the vehicle reaches the stability limit or not, the additional yaw moment is applied. In addition, there are motion reference models in both the LSC layer and the TTC layer that are used to predict the vehicle motion state. The use of different reference models for the two may have unexpected effects on the stability-tracking control system.

To investigate the influence of reference models on the control results of the system in the LSC layer, five reference models are established in this section, namely Linear 2-DOF Model (L2-DOFM), First-Order Steady State Gain Model (1-OSSGM), Neutral Steering Steady State Gain Model (NS-SSGM), Understeering Steering Steady State Gain Model (US-SSGM) and Oversteering Steering Steady State Gain Model (OS-SSGM). According to whether there is a dynamic response process, the five reference models can be divided into the model with dynamic response and the steady-state gain model. These models are used here for comparative experiments.

#### 3.3.1. Steady-State Gain Model

The steady-state gain is as follows:

$$\gamma_{d\_SSGM} = \frac{v_x}{l(1 + Kv_x^2)}\delta_f \tag{18}$$

where

$$K = \frac{m}{l^2}\left(\frac{l_f}{k_r} - \frac{l_r}{k_f}\right) \tag{19}$$

When $K > 0$, the vehicle has understeering characteristics. When $K < 0$, the vehicle has oversteering characteristics. When $K = 0$, the vehicle has neutral steering characteristics. By setting $K$ value, a reference model with understeering, oversteering and neutral steering characteristics can be obtained. The original vehicle using the parameters in Table 1 has a minimal stability factor value, and it has a neutral steering characteristics.

#### 3.3.2. Dynamic Response Model

L2-DOFM is established as follows:

$$\begin{bmatrix} a_y \\ \dot{\gamma} \end{bmatrix} = \begin{bmatrix} \frac{-(k_f+k_r)}{mv_x} & \frac{l_fk_r-l_rk_f}{mv_x} - v_x \\ \frac{l_rk_r-l_fk_f}{I_zv_x} & \frac{-(l_f{}^2k_f+l_r{}^2k_r)}{I_zv_x} \end{bmatrix} \begin{bmatrix} v_y \\ \gamma \end{bmatrix} + \begin{bmatrix} \frac{k_f}{m} \\ \frac{l_fk_f}{I_z} \end{bmatrix} \begin{bmatrix} \delta_f \end{bmatrix} \tag{20}$$

The yaw rate of the vehicle at t + TPS is predicted as follows:

$$\gamma_{d\_L2-DOFM}(t + T_{PS}) = C_\gamma \cdot e^{A_S \cdot T_{PS}} \cdot X(t) + C_\gamma \cdot \int_0^{T_{PS}} e^{A_S\eta}d\eta B_S u \tag{21}$$

where $T_{PS}$ is the predicted time, $C_\gamma = [0, 1]$, $A_S \in R2 \times 2$ and $B_S \in R2 \times 1$ are the coefficient matrix of state vector $[v_y\ r]^T$ and control vector $[\delta_f]$ in Formula (21) respectively. The L2-DOFM is the same as the reference model in the trajectory tracker.

Moreover, 1-OSSGM is established as follows:

$$\gamma_{d\_1-OSSGM}(s) = \frac{v_x}{l(1 + Kv_x^2)} \cdot \frac{1}{\tau s + 1}\delta_f \tag{22}$$

where $\tau$ is the time constant, which represents the transient response speed of the reference model. In fact, 1-OSSGM is the product of NS-SSGM and a first-order inertia element. Figure 8 shows the unit step response of 1-OSSGM and L2-DOFM.

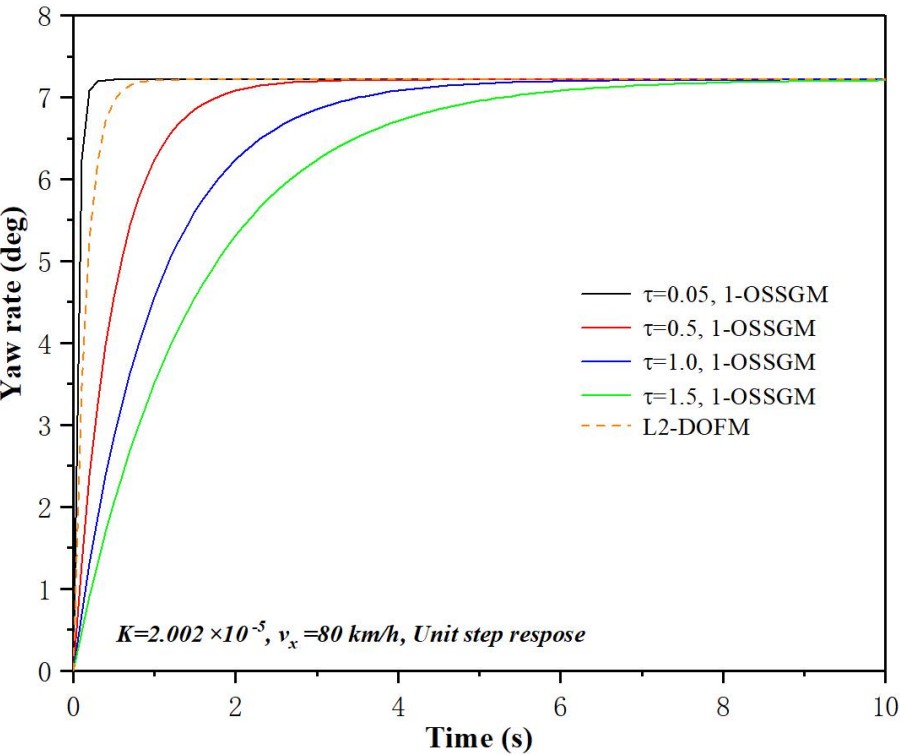

**Figure 8.** Angular step response.

The larger the time constant of 1-OSSGM, the slower the steady-state value is reached. By adjusting the time constant, it is found that the unit step response curves of 1-OSSGM and L2-DOFM can approximately coincide when the time constant $\tau$ is set between 0.05 and 0.5. It can be inferred that an infinitely small time constant, $\tau$, would cause the dynamic response process of the reference model to vanish.

The deviation of the vehicle state is as follows:

$$\Delta e_{ms} = \gamma_d - \gamma \tag{23}$$

The additional yaw moment can be calculated by using the PID algorithm:

$$\Delta M_Z = K_S[\Delta e_{ms} + \frac{1}{T_i}\int_0^t \Delta e_{ms}dt + T_d\frac{d\Delta e_{ms}}{dt}] \tag{24}$$

where $K_S$, $T_i$ and $T_d$ are proportional coefficient, integral time and differential time, respectively. $K_S$ also represents the stability correction strength, that is, which is the ratio coefficient of the yaw moment to the deviation of the vehicle state. The stability correction strength is the most important parameter of the LSC layer.

For DDEVs equipped with redundant actuators, allocation control is one of the key technologies. The methods are divided into rule-based torque allocation methods and torque allocation methods based on optimization theory [26,30]. The former establishes proportional relationships for torque allocation between different sides and different wheels, and it has a simple implementation and low computational effort. The latter fully takes into account the actuator operating limits and the friction efficiency of the road and tires, and it facilitates the full use of distributed drive vehicle stability control advantages. Considering that the front axle tire responsible for steering will consume part of the adhesion force, in

order to make the tire as much as possible to obtain road adhesion force, the minimizing tire utilization rule based on the axial load proportional distribution is proposed.

The objective function is established as follows:

$$J = \min \sum_{i=1}^{4} \frac{T_{xi}^2}{(\mu F_{zi} R)^2} \tag{25}$$

The constraint conditions are as follows:

$$s.t. \begin{cases} \left(T_{xfl} + T_{xfr}\right) \cos\left(\delta_f\right) + T_{xrl} + T_{xrr} = T_x \\ \frac{B}{2R}\left(T_{xfr} - T_{xfl}\right) \cos\left(\delta_f\right) + \frac{B}{2R}\left(T_{xrr} - T_{xrl}\right) = \Delta M_z \\ T_{xfl} + T_{xfr} = p \cdot (T_{xrl} + T_{xrr}), -1 \leq p \leq 1 \\ |T_{xi}| \leq \min|(\mu_i F_{zi} R, T_{\max})| \end{cases} \tag{26}$$

where the third term is the constraint of proportional distribution of axle load, and $p$ is the distribution coefficient of front and rear axle load. The distribution method fully considered the constraints of the adhesion utilization of each tire and maximum torque provided by the motor. The torque distribution problem was transformed into a problem of solving quadratic programming.

The control process of the LSC layer is summarized in three parts: vehicle state prediction, vehicle motion control and tire torque distribution, as shown in Figure 9. As can be seen from Figure 10 and Formulas (18)–(26), the main parameters that determine the output results are the same as for the trajectory tracking controller except for the stability correction strength. According to the coupling relationship of vehicle dynamics, these parameters change in real time except the predicted time. These changes will affect the calculation results of front wheel steer angle and braking/driving force. As the input of the lateral stability controller, the front wheel steer angle determines the vehicle reference motion state value and the vehicle motion state in a period of time in the future. As the input of the LSC layer, the drive/braking forces determines the sum of the force in the torque distribution layer, which indirectly affects the torque distribution of each wheel.

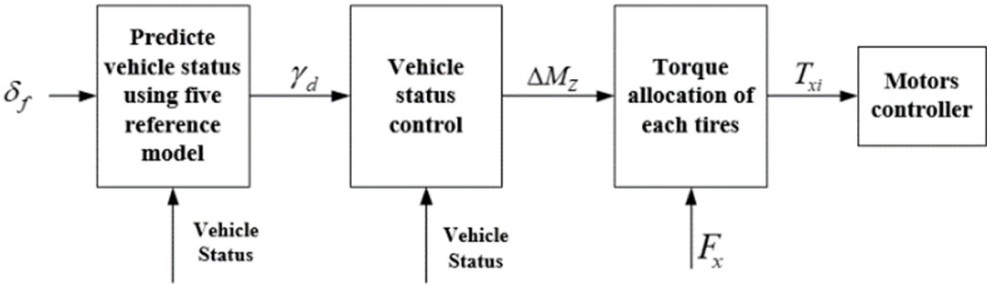

**Figure 9.** Lateral stability control layer.

*3.4. An Intrinsic Mechanistic Framework for Stability-Tracking Control*

Comparing the control logic of the LSC layer with that of the TTC layer, it can be found that the former exerts its effect on the latter in two ways: one is through the front wheel steer angle and the braking/driving force directly affect the calculation of reference yaw rate and additional yaw moment, the second is through the state feedback from the vehicle dynamics coupling motion indirectly. The LSC layer can only exert an indirect effect on the TTC through the second way. Therefore, the interaction relationship between the two layers can be expressed as follows: the trajectory tracking controller plays an active role, and the lateral stability controller plays a follower role.

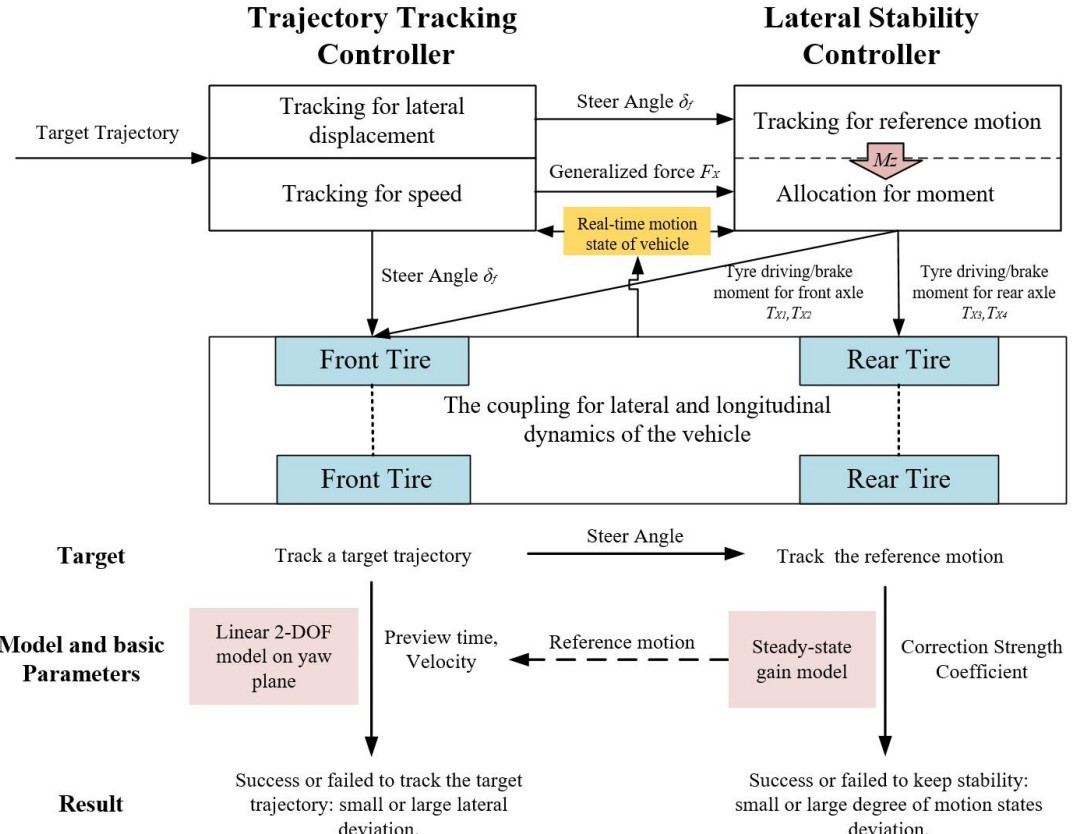

**Figure 10.** Intrinsic mechanistic framework for stability-tracking control.

Summarizing the above modeling process and analysis, an intrinsic mechanistic framework for stability-tracking control is proposed, as shown in Figure 10. This framework describes the general principles of TTC and LSC when a hierarchical mechanism is used, including the control flow, control objective, reference model, control parameters and control results of the two controllers, and the relationship between them is represented by arrows. The real-time motion state of vehicles needs to be realized by some estimation algorithms, which are simplified in this paper.

According to the framework, we made the following analysis:

From the perspective of control flow, the outputs of both layers are involved in the coupling relationship between lateral dynamics and longitudinal dynamics of the vehicle. The TTC layer is involved in lateral dynamics by controlling the front wheel steer angle, while the LSC layer is involved in longitudinal and lateral dynamics by applying different driving/braking torques to each wheel. The lateral force and moment for the vehicle will cause changes in longitudinal and lateral movement, as well as changes in roll and yaw movement. These varying vehicle states are transmitted to the TTC layer and the LSC layer through feedback channels, and they affect the results of the STC system.

From the perspective of control objectives, the two control layer have different control objectives, namely tracking reference motion state and tracking target trajectory. In the process of tracking the target trajectory, the sideslip angle of the tires becomes larger due to the large change of and the rate of the amplitude of the front wheel steer angle; the tire will easily enter the nonlinear region and affect the stability of the vehicle. In the process of tracking reference motion, the additional yaw moment will make the actual motion state of the vehicle consistent with the reference motion state, and this will have an unpredictable effect on the trajectory prediction of the TTC layer. The interaction between the two control layer may be favorable or unfavorable to the results of stable-tracking control. If the control objective of stability controller and trajectory tracking controller conflicts, the control effect of the control system may become worse.

From the perspective of basic parameters, in the closed-loop control process, the change of parameters will affect the performance of the whole control system. The main parameters of STC system include preview time, speed and stability correction strength. According to the analysis of the relationship between the predicted time ($T_{PT}$) and $G$ in the previous chapter, changes in the preview time and vehicle speed will change the amplitude and rate of change of the steering input. Turning too fast will change the tire sideslip angle, the sideslip angle and the yaw rate, which will affect the stability of the vehicle. By adjusting the strength coefficient of stability correction, the degree of approaching to the reference motion of the vehicle is changed and the trajectory of the vehicle is affected. In addition, the target trajectory is the input of the stability-tracking control system, and the change of curvature and road adhesion conditions will affect the steer angle of the TTC layer and the torque distribution.

Based on the proposed framework and the above analysis, it can be inferred that the key to understanding the intrinsic mechanistic of the interaction of the two control layers is to determine the control target, control parameters, reference model and the law of the target trajectory, affecting the performance of the stability and trajectory tracking.

## 4. Simulation Results and Analysis

In order to investigate the intrinsic mechanism of stability-tracking control and to obtain the best control results, this paper quantitatively and qualitatively investigates the effects of parameters, two control objectives and reference models on the STC system by designing simulation experiments. In order to evaluate the control effect of the STC system, the area and peak value of the phase plane portrait of yaw velocity and sideslip angle are used to evaluate the vehicle stability performance, and the peak value of the lateral displacement deviation of the vehicle tracking target trajectory is used to evaluate the tracking effect. Table 2 shows the unchanged controller parameters.

**Table 2.** Unchanged controller parameters.

| Parameters | $T_{iT}$ | $T_{dT}$ | $K_T$ | $T_i$ | $T_d$ | $p$ |
|---|---|---|---|---|---|---|
| Value | 4 | 0.05 | 800 | 2 | 0.1 | 0.7 |

### 4.1. Influence of Target Trajectory on Control System

The road adhesion coefficient and curvature are the basic characteristics of target trajectory. The greater the adhesion coefficient, the better the tire adhesion ability. The adhesion ability of the tire can be represented by the adhesion ellipse, which is the vector sum of the longitudinal force and the lateral force of the tire [30,31]. According to the proposed framework, the adhesion coefficient makes the two controllers unable to achieve their respective control objectives by affecting the adhesive force. The curvature refers to the reciprocal of the radius of the curve trajectory. Its influence on the stability-tracking control system is unclear.

In order to study the influence of curvature on the performance of trajectory tracking and stability, three right-angle curves with different curvature are designed as target trajectories in this section, as shown in Figure 11. Table 3 shows the lateral displacement deviation under right-angle bend with different radius.

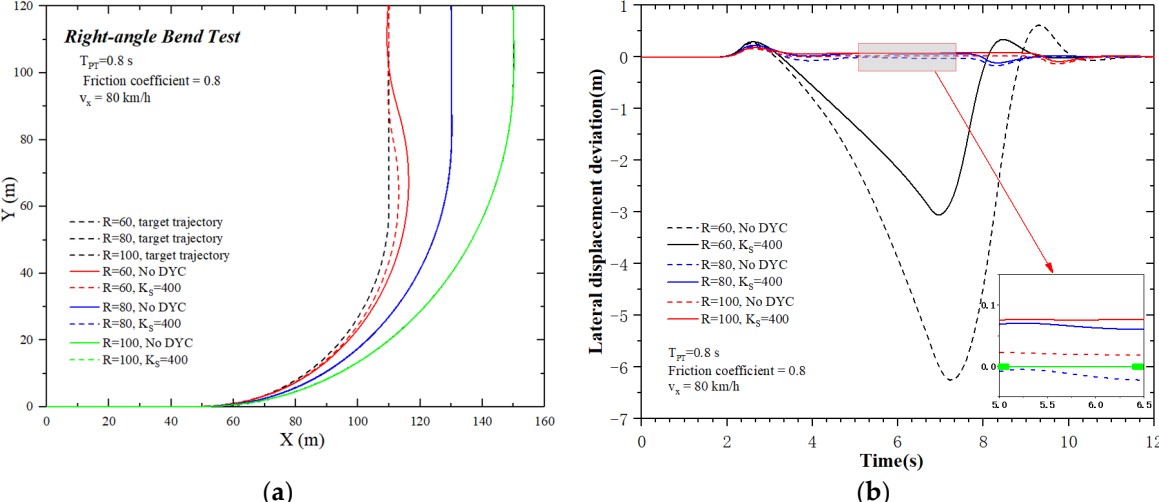

**Figure 11.** Trajectory tracking performance under different road curvature: (**a**) trajectory and (**b**) lateral displacement deviation.

**Table 3.** Lateral displacement deviation under different right-angle bend with different radius.

| Peak Values | DYC | R = 60 m | R = 80 m | R = 100 m |
|---|---|---|---|---|
| $\Delta Y_{max}$ (m) | $K_S = 0$ (No DYC) | 6.26 | 0.02 | 0.03 |
| | $K_S = 400$ | 3.06 | 0.07 | 0.08 |

When the curvature is large, the yaw moment can effectively reduce the sideslip degree of the vehicle, the lateral displacement deviation $\Delta Y_{max}$ will be reduced and the track tracking effect will be improved. However, when the curvature is large, although the vehicle can track the target trajectory well both when the yaw moment is applied ($K_S = 400$) and when the yaw moment is not applied ($K_S = 0$), and the lateral displacement deviation of the former is greater than that of the latter, and its track tracking performance is worse than that of the latter. Therefore, there is a contradiction between them when the curvature is small, and the influence of LSC on trajectory tracking is different when the curvature of target trajectory is different.

As can be seen from Figure 12, compared with the results of applying DYC, when the curvature is small, the area covered by the phase plane curve of yaw rate and sideslip angle without DYC is smaller, and the peak value of yaw rate and beta are both smaller. However, when the curvature is larger, the area covered by the phase plane curve of the latter is larger, and the peaks of the yaw rate and the sideslip angle are also larger. This is inconsistent with common perception.

The reason for this phenomenon is that the objective of LSC layer conflicts with that of TTC. When the actual trajectory of the vehicle coincides with the target trajectory, the motion mode of the vehicle tracking the right-angle curve trajectory is approximately uniform circular motion. According to the uniform circular motion formula of a particle, the lateral acceleration and yaw rate of the vehicle increase with the increase of curvature at the same speed. For example, when the vehicle speed is 80 km/h and the right-angle turning radius is 100 m, the corresponding lateral acceleration is 0.504 g, which is approximately equal to the steady-state region ($t = 4\sim9$ s) of the curve in Figure 12b. Therefore, the closer the actual trajectory of the vehicle is to the target trajectory, the closer its yaw rate and lateral acceleration are to the yaw rate and lateral acceleration corresponding to the target trajectory in uniform circular motion. When the target trajectory curvature is enough large, the lateral acceleration of the vehicle centroid will increase, and the lateral force of the tire is easier to reach saturation gradually. At this time, if the vehicle tracks the corresponding reference motion state by using DYC, the vehicle may be more prone to instability. On the

other hand, if the steering wheel angle is reduced to reduce the lateral acceleration, the TTC effect will become worse.

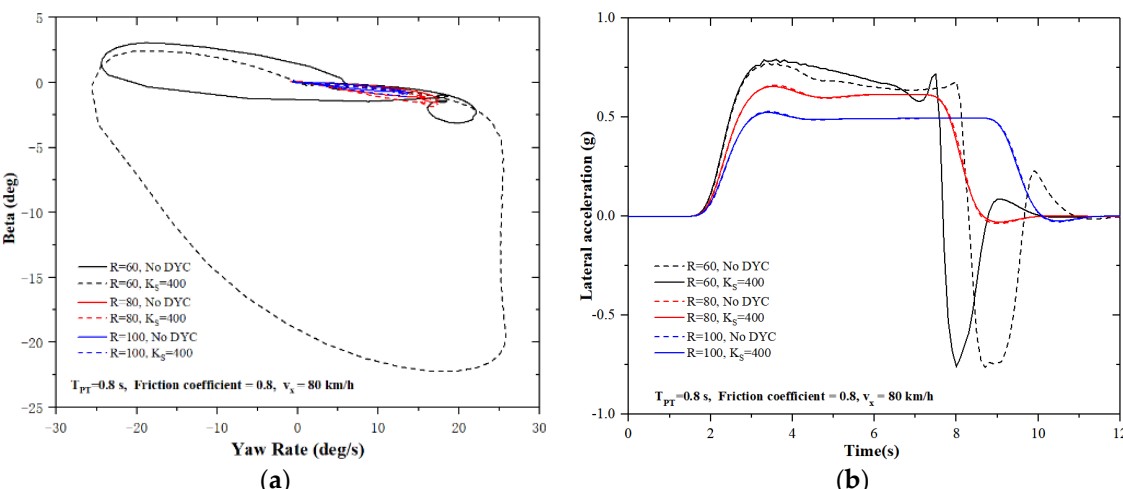

**Figure 12.** Stability performance under different road curvature: (**a**) phase plane portrait of yaw velocity and sideslip angle; (**b**) lateral acceleration.

Therefore, when tracking target trajectories with different curvatures, some trade-offs should be made between the two targets.

### 4.2. Influence of Reference Model on Control System

In this section, double lane change trajectory is used as the target trajectory to study the influence of five vehicle reference models on vehicle stability and trajectory tracking effect.

Firstly, by comparing steady-state gain models NS-SSGM, OS-SSGM and US-SSGM with L2-DOFM, it can be seen from Figure 13 that the time constant, $\tau$, of 1-OSSGM has a great influence on the control effect. Table 4 shows the lateral displacement deviation and maximum steering wheel angle with different reference models. The trajectory curves of $\tau = 0.5$ and $\tau = 1.0$ are obviously worse than those of L2-DOFDM at 110 and 150 m lateral displacements, and the latter curve has obvious fluctuation at the end of the target trajectory. In addition, the delay can also cause the steering wheel to turn at a greater angle and jarring. Moreover, it can be inferred from Figure 8 that the lag effect of 1-OSSGM's dynamic response process increases gradually with the increase of $\tau$. This feature makes the direction control and DYC appear to be delayed, and the steering wheel angle and vehicle motion state cannot be corrected in time.

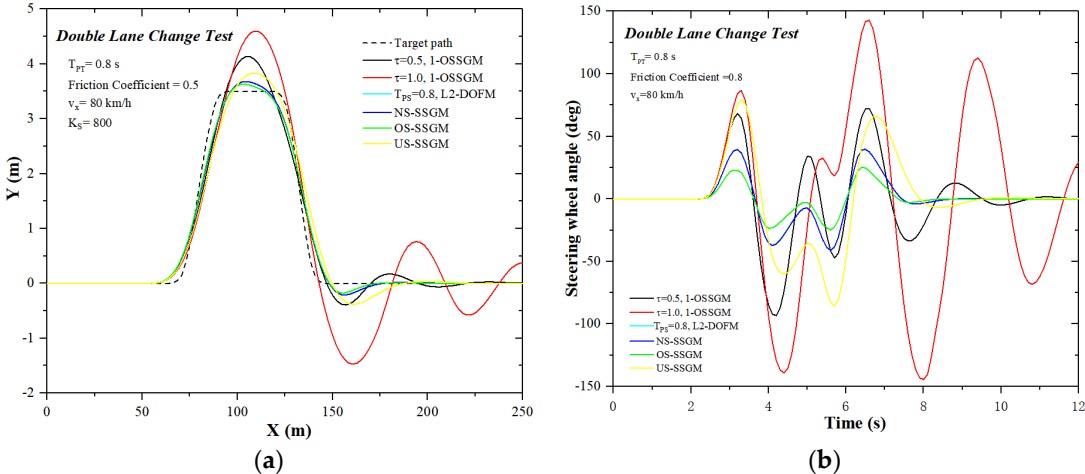

**Figure 13.** Trajectory tracking performance under different reference models: (**a**) trajectory and (**b**) steering wheel angle.

**Table 4.** Lateral displacement deviation and maximum steering wheel angle with different reference models.

| Peak Values | 1-OSSGM, $\tau = 0.5$ | 1-OSSGM, $\tau = 1$ | L2-DOFM, $T_{PS} = 0.8$ s | NS-SSGM | OS-SSGM | US-SSGM |
|---|---|---|---|---|---|---|
| $\Delta Y_{max}$ (m), near X = 100 | 0.63 | 1.10 | 0.18 | 0.18 | 0.13 | 0.32 |
| $\Delta Y_{max}$ (m), near X = 150 | 0.38 | 1.47 | 0.21 | 0.21 | 0.17 | 0.39 |
| $\delta_{max}$ (deg) | 93.3 | 143.9 | 40.7 | 40.7 | 25.4 | 84.8 |

NS-SSGM can be used to obtain the reference value of yaw rate without delay, and L2-DOFM can be used to obtain the reference value of yaw rate with small delay time. It can be seen from Figure 13 that the trajectory curves of L2-DOFM and NS-SSGM coincide. The reason for the coincidence is that, when the predicted time, $T_{PS}$, of the lateral stability controller is long enough, the vehicle motion state has reached the steady state, and the predicted result of L2-DOFDM is the same as that of the steady gain model NS-SSGM. The steady-state value of the unit step response of L2-DOFDM in Figure 8, which has reached 90% at 0.36 s, can confirm the above views. Therefore, when the predicted time TPS is long enough, L2-DOFDM and the original steady-state gain model are used as the reference model; it has little influence on the stable-tracking control system.

Then, by comparing the steady-state gain models NS-SSGM, OS-SSGM and US-SSGM, it can be found that the trajectory of the three models show their respective steering characteristics, and the vehicle reference model with different steering characteristics will also affect the effect of TTC and the amplitude of steer angle input. US-SSGM shows a large overshoot at the maximum lateral offset and the end of the double lane change trajectory, while OS-SSGM shows the smallest overshoot. Furthermore, the use of OS-SSGM results in smaller steering wheel angles. The results show that the steering characteristics of vehicles can be changed by DYC to influence the TTC.

Figure 14 shows a phase plane portrait of yaw velocity and sideslip angle under different reference models. It can be found that the delay effect and steering characteristics of vehicle reference model both affect vehicle stability. The larger the delay is, namely $\tau$, and the closer the steering characteristics are to the tendency of oversteering, the larger the area enclosed by the phase locus of yaw velocity and sideslip angle, and the worse the vehicle stability will be.

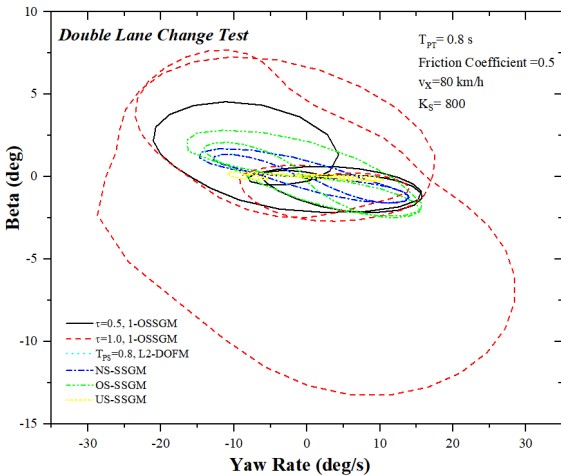

**Figure 14.** Phase plane portrait of yaw velocity and sideslip angle under different reference models.

On the contrary, the closer the steady-state steering characteristic is to the understeering, the worse the trajectory tracking performance may become. Therefore, the steering characteristics and delay effects of the vehicle reference model should be fully utilized to obtain the best trajectory tracking performance and vehicle stability.

### 4.3. Influence of Preview Time on the Control System

Five groups with different preview time were adopted, and take the double lane change trajectory as the target trajectory, the simulation experiment is carried out.

As can be seen from Figure 15, the greater the preview time is, the flatter the curvature of the actual trajectory of the vehicle is, and the earlier the trajectory of the vehicle deviates from the target trajectory is, and the larger the overshoot at the lateral displacement of about 100 and 150 m and at the end of the curve is. Table 5 shows the lateral displacement deviation and maximum steering wheel angle under different preview time. In addition, both peak steering wheel angle and change rate decrease with the increase of preview time. It is observed that the overshoot occurs when the sign of the course angle of the path changes. Before the overshoot occurs, the closer the actual trajectory of the vehicle is to the target trajectory, the greater the overshoot will be. In order to improve the tracking ability, it is necessary to balance the overshoot and the proximity between the actual trajectory and the target trajectory. Preview time $T_{PT}$ can mediate both. The essence of this method lies in the regulatory effect of $T_{PT}$ on parameter G mentioned in the previous chapter.

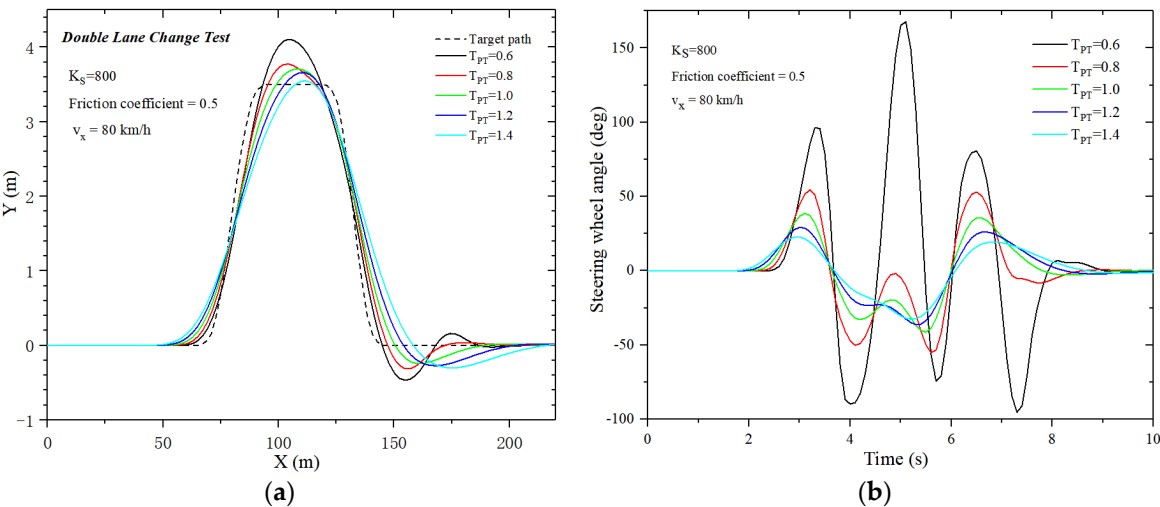

**Figure 15.** Trajectory tracking performance under different predicted time $T_{PT}$: (**a**) phase plane portrait of yaw velocity and sideslip angle; (**b**) steering wheel angle.

**Table 5.** Lateral displacement deviation and maximum steering wheel angle under different preview time.

| Peak Values | $T_{PT}$ = 0.6 s | $T_{PT}$ = 0.8 s | $T_{PT}$ = 1.0 s | $T_{PT}$ = 1.2 s | $T_{PT}$ = 1.4 s |
|---|---|---|---|---|---|
| $\Delta Y_{max}$ (m), near X = 100 | 0.61 | 0.28 | 0.21 | 0.16 | 0.05 |
| $\Delta Y_{max}$ (m), near X = 150 | 0.47 | 0.31 | 0.24 | 0.26 | 0.31 |
| $\delta_{max}$ (deg) | 166.9 | 54.8 | 41.1 | 36.4 | 32.1 |

The regulatory effect of $T_{PT}$ on $G$ also affects vehicle stability. As can be seen from Figure 16, when the preview time, $T_{PT}$, keeps increasing, the area enclosed by the phase locus of yaw rate and sideslip angle gradually decreases, and the vehicle performance becomes more stable. The smaller $T_{PT}$ will cause the larger sideslip angle and yaw rate, which will easily lead to vehicle instability. The reasons for instability are that a smaller $G$ value can be calculated with a smaller predicted time, $T_{PT}$, and a smaller $G$ value will increase the increment of steering wheel angle, and the larger change and rate of steering wheel angle will cause the vehicle's motion state to approach the instability boundaries. At this point, if the vehicle tracks the motion state of the reference model, the actual motion state of the vehicle will be closer to the instability boundaries, which is contrary to the goal of LSC. In some of the literature [15,32], thresholds are often set to prevent vehicles

from experiencing extreme yaw rate. However, the threshold may cause the motion state to fail to reach the expectation in the process of trajectory tracking, weaken the steering ability and ultimately make the trajectory tracking effect of the vehicle worse. Therefore, the effect of preview time on vehicle stability should be fully considered when selecting LSC parameters and setting the thresholds.

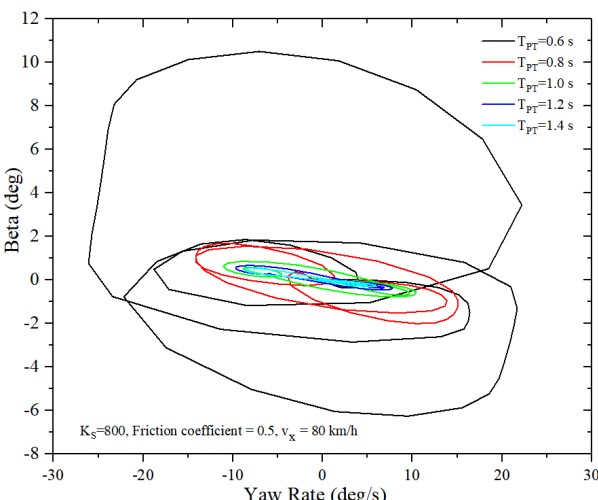

**Figure 16.** Phase plane portrait of yaw velocity and sideslip angle under different preview time, $T_{PT}$.

### 4.4. Influence of Stability Correction Coefficient on the Control System

Six groups of different stability correction coefficient, $K_S$, were used to carry out simulation experiments, and double lane change trajectory is used as target trajectories.

As can be seen from Figure 17, adjusting $K_S$ can reduce the proximity between vehicle trajectory and target trajectory and the overshoot, but the change is not as obvious as that caused by adjusting the $T_{PT}$. Table 6 shows the lateral displacement deviation and maximum steering wheel angle under different stability correction strength. In addition, the peak value of the steering wheel angle will be reduced. According to Formula (3), the additional yaw moment through the difference of the longitudinal force of the tires and the yaw moment generated by lateral force of the tires through the change of the front wheel steer angle together constitute the power of vehicle yaw motion. When the proportion of the former increases gradually, the vehicle motion state will be closer to the reference motion state, and the latter will be more efficient and the peak value of the front wheel steer angle will be reduced. However, when the motion state of the vehicle is close enough to the reference motion state, the influence of the former is gradually weakened. As shown in Figure 17b, as $K_S$ gradually increases, the peak value of steering wheel angles no longer decreases. Moreover, the predicted results of the TTC layer become more accurate, and the trajectory tracking effect of the vehicle is also improved to some extent. As shown in Figure 17a, the overshoot of the two enlarged graphs reaches the minimum of $K_S = 400 \sim 800$ to illustrate this point.

Moreover, the trajectory tracking controller plays a major role in the yaw motion of the vehicle, and the yaw moment output by the lateral stability controller plays a secondary role in the yaw motion of the vehicle. Tracking the reference motion state is the sole control objective of the lateral stability controller, and it is obtained by the reference model. The front wheel steer angle is the output of the trajectory tracking controller and the input of the lateral stability controller, which is related to the curvature change of the trajectory and the vehicle speed. Therefore, the trajectory tracking controller will directly affect the control target of the lateral stability controller. As shown in Figure 18, when the stability correction coefficient, $K_S$, increases, the longitudinal force on the wheel will increase, and the phase plane curve of yaw velocity and sideslip angle is the comprehensive result of yaw moment generated by the trajectory tracking controller and the stability controller. At

this time, the road adhesion condition is good, the sideslip degree of the vehicle is low and the vehicle stability is good, but the peak value of the sideslip angle and yaw velocity of the vehicle are not decrease, or even increase. This corresponds to the study of the influence of curvature on vehicle stability in Section 4.1.

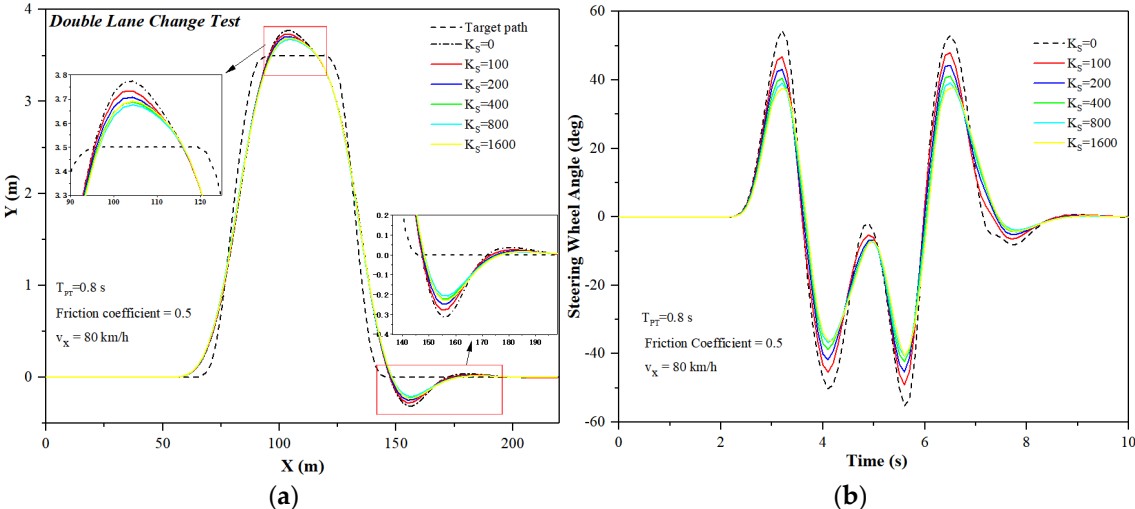

(a)                                                                    (b)

**Figure 17.** Trajectory tracking performance under different stability correction coefficient: (**a**) trajectory and (**b**) steering wheel angle.

**Table 6.** Lateral displacement deviation and maximum steering wheel angle under different stability correction strength.

| Peak Values | $K_S$ = 0 (NO DYC) | $K_S$ = 100 | $K_S$ = 200 | $K_S$ = 400 | $K_S$ = 800 | $K_S$ = 1600 |
|---|---|---|---|---|---|---|
| $\Delta Y_{max}$ (m), near X = 100 | 0.27 | 0.23 | 0.20 | 0.18 | 0.17 | 0.19 |
| $\Delta Y_{max}$ (m), near X = 150 | 0.31 | 0.27 | 0.24 | 0.22 | 0.20 | 0.23 |
| $\delta_{max}$ (deg) | 55.1 | 48.9 | 44.8 | 42.4 | 40.6 | 40.7 |

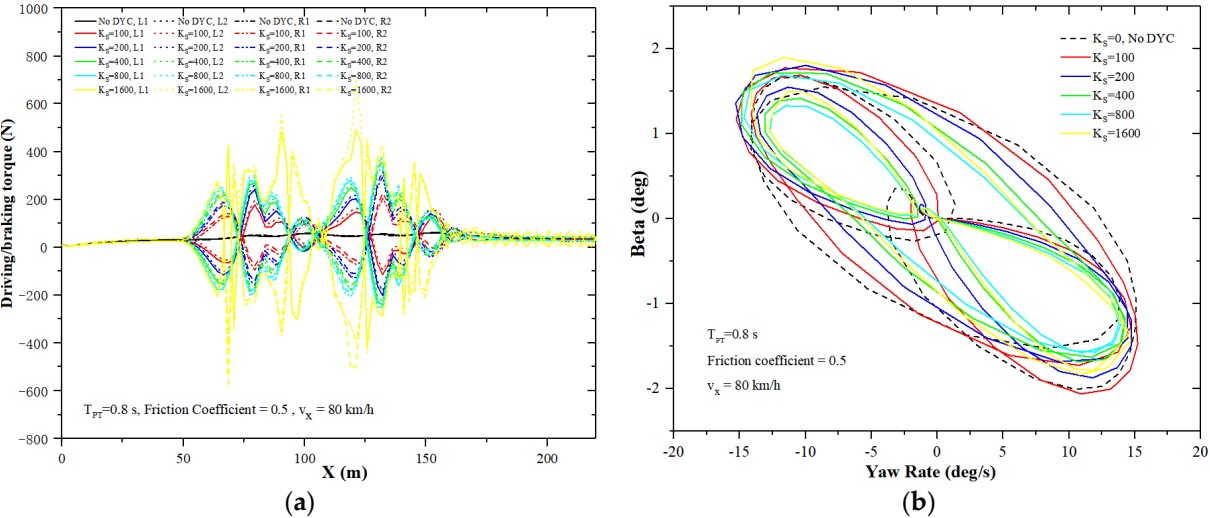

(a)                                                                    (b)

**Figure 18.** Vehicle stability performance under different stability correction coefficient: (**a**) driving/braking torque; (**b**) phase plane portrait of yaw velocity and sideslip angle.

## 5. Conclusions

The intrinsic mechanism framework of the stability-tracking control proposed in this paper explains the reasons for the interaction between trajectory tracking control and lateral stability control. The process of their interaction was analyzed by establishing the vehicle dynamics relationship and trajectory tracking control layer and lateral stability control layer. The interaction between the two was investigated in three aspects, namely target trajectory, reference model and parameters, through simulation experiments.

In the simulation experiments, some of the most valuable laws were obtained: the curvature of the target trajectory leads to the conflict between two control objectives during the trajectory tracking, the trajectory tracking and lateral stability performance of the vehicle under the reference model with different steering characteristics and delay characteristics, the regulatory effects of preview time and stability correction strength on the trajectory tracking and lateral stability performance of the vehicle.

**Author Contributions:** Conceptualization, X.T., Y.Y. and L.Z.; methodology, X.T. and Y.Y.; software, X.T.; validation, X.T.; formal analysis, X.T. and B.W.; investigation, X.T., B.W., X.X. and L.Z.; resources, X.T. and Y.Y.; writing—original draft preparation, X.T., Y.Y. and B.W.; writing—review and editing, X.T., Y.Y., X.X., L.Z. and B.W.; supervision, Y.Y. and B.W.; funding acquisition, Y.Y. and X.X. All authors have read and agreed to the published version of the manuscript.

**Funding:** This research was funded by the National Natural Science Foundation of China (51975428 and 51975426).

**Acknowledgments:** The authors would like to thank the National Natural Science Foundation of China (51975428 and 51975426).

**Conflicts of Interest:** The authors declare no conflict of interest.

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
