# Peer review of "Analysis of Intrinsic Mechanistic of Stability-Tracking Control for Distributed Drive Autonomous Electric Vehicle"

_electronics, doi:10.3390/electronics10233010_

Round 1

Reviewer 1 Report

The emergence of electrical vehicles has opened the opportunity for control techniques that were not previously possible. One example is distributed drive control, where multiple electrical motors can be used in one vehicle. This manuscript considers the distributed derive paradigm in electrical vehicles, where two previously independent control mechanisms, I.e., lateral stability control and trajectory control, are ruled into one overarching controller. In principal, the authors explain the derived control well and the concept is supported via simulations. I can recommend the manuscript for publication.    

Author Response

Dear reviewer:

Thank you for your comments and affirmation concerning our manuscript.

Your kind considerations will be greatly appreciated.

With best regards,

Sincerely Yours,

Mr. Xuequan Tang

Reviewer 2 Report

The manuscript presents trajectory tracking control method with lateral stability control. These methods are applied to autonomous electric vehicle. The main problem which I see in the proposed manuscript is that the topic of the paper does not correspond to the subject of the Electronics Journal. Inside of the paper reader cannot find anything which would involve electronics. I recommend submitting your manuscript to another journal from the filed of Mechanics, Vehicles or Physics. 

Author Response

Dear reviewer:

Thank you for your comments concerning the manuscript.

For the comment that “The manuscript presents trajectory tracking control method with lateral stability control. These methods are applied to autonomous electric vehicles. The main problem which I see in the proposed manuscript is that the topic of the paper does not correspond to the subject of the Electronics Journal. Inside of the paper reader cannot find anything which would involve electronics. I recommend submitting your manuscript to another journal from the filed of Mechanics, Vehicles or Physics. ”, we try our best to answer your comments from the following aspects:

1. Subject areas of Electronics Journal

We browsed the Subject areas of Electronics Journal and found that the two related to this article are Systems & Control Engineering and Electrical and Autonomous Vehicles. Distributed drive electric vehicles (DDEVs) and autonomous vehicles using pure electric chassis are the main topic in our manuscript. Trajectory tracking control is one of the key technologies for autonomous driving. In the distribution method, the maximum torque provided by motors was considered as the constraints. These parameters are derived from real motors, which affect the torque distribution of each wheel. The specific details can be found in the text.

2. Published papers

We have counted several related papers published in Electronics Journal and other journals in recent years as follows:

[1] Nah J, Yim S. Vehicle Stability Control with Four-Wheel Independent Braking, Drive and Steering on In-Wheel Motor-Driven Electric Vehicles. Electronics. 2020 Nov;9(11).

[2] Chen J, Shuai Z, Zhang H, et al. Path Following Control of Autonomous Four-Wheel-Independent-Drive Electric Vehicles via Second-Order Sliding Mode and Nonlinear Disturbance Observer Techniques. Ieee Transactions on Industrial Electronics. 2021 Mar;68(3):2460-2469.

[3] Ding S, Liu L, Zheng WX. Sliding Mode Direct Yaw-Moment Control Design for In-Wheel Electric Vehicles. Ieee Transactions on Industrial Electronics. 2017 Aug;64(8):6752-6762.

[4] Dizqah AM, Lenzo B, Sorniotti A, et al. A Fast and Parametric Torque Distribution Strategy for Four-Wheel-Drive Energy-Efficient Electric Vehicles. Ieee Transactions on Industrial Electronics. 2016 Jul;63(7):4367-4376.

Our research attempts to analyze the mechanism of the interaction between the two controls and provide certain guidance for the design of the two controllers. The torque distribution method of the distributed drive chassis belongs to an indispensable research hotspot in electrification research.

3. Extensive Interests:

Many scholars who are engaged in the coordinated control of path-following and lateral stability research for DDEVs, will be very interested in this. Research on this issue has the following significance:

(1) by analyzing whether the lateral stability control (LSC) to the trajectory tracking control (TTC) will conflict under their respective control objectives, guidance can be put forward for the stability-tracking coordination control method;

(2) By analyzing the process of STC, the main factors affecting STC can be found, and the efficiency of STC system can be improved.

In addition, we count the number of papers published about theme in the past three years. Starting in 2019, the number of related papers is 2, 3, and 7 respectively. These data are retrieved from Web of Science. These articles all mention our subject clearly in the title.These data may fluctuate, but the trend shows that this topic is getting more and more attention from scholars.

We firmly believe that our contribution can provide some new references and valuable conclusions for their research. They can more easily get this paper from this journal.

We sincerely hope that our above explanation can be adopted by you.

Thank you again for your comments on our manuscript. We will make appropriate revisions to the manuscript based on your comments. Please see the attachment for the revised manuscript.

With best regards,

Sincerely Yours,

Mr. Xuequan Tang

Reviewer 3 Report

I think the content of this paper is written on a very interesting topic.

This study was written on the premise that the effect of LSC addition on the TTC system is not yet clear.

Therefore, the authors first built a Stability-Track hierarchical control structure composed of LSC and TTC and identified the interaction between the two layers as the core of this paper. And the authors differentiated proposing a unique mechanism framework of stability tracking control (STC) by establishing and analyzing two-layer vehicle dynamic model and control process.

There are several enhancements.
1. It was described that the interaction process was analyzed by setting the vehicle dynamics relationship, the trajectory tracking control layer, and the lateral stability control.
: There are some insufficient parts to understand the related analysis results in the text, and I request that you compare the results such as quantitative ratios in a separate table. I wish the size of the picture could be improved.

2. Some of the most valuable laws were obtained in simulation experiments. The curvature of the target trajectory leads to a collision between the two control targets during trajectory tracking. In this expression, please emphasize the characteristics obtained through simulation analysis results as a comparative numerical value and comparison.
3. It would be nice to add information about the trajectory tracking controller and the stability controller.

Author Response

Dear reviewer:

Thank you for your comments and affirmation concerning our manuscript.

We have tried our best to improve the manuscript.

Reviewer’s comments 1:  It was described that the interaction process was analyzed by setting the vehicle dynamics relationship, the trajectory tracking control layer, and the lateral stability control.There are some insufficient parts to understand the related analysis results in the text, and I request that you compare the results such as quantitative ratios in a separate table. I wish the size of the picture could be improved.

Response: We appreciate it very much for this good suggestion, and we have done it according to your ideas. In the section of simulation results and analysis, four tables (Table 3, Table 4, Table 5 and Table 6) have been added, and the pictures have been enlarged so that readers and reviewers can see more clearly.

Reviewer’s comments 2: Some of the most valuable laws were obtained in simulation experiments. The curvature of the target trajectory leads to a collision between the two control targets during trajectory tracking. In this expression, please emphasize the characteristics obtained through simulation analysis results as a comparative numerical value and comparison.

Response: We extracted the important numerical information in the curve and put it into four tables. The comparison of these numerical values greatly facilitated the interpretation of the laws we proposed. This expression is more convincing. We think your comments have improved our manuscript. Thank you very much.

Reviewer’s comments 3: It would be nice to add information about the trajectory tracking controller and the stability controller.

Response: In response to this comment, we added Table 2 to indicate the unchanged controller parameters. Other parameters, such as KS, preview time TPT, vehicle speed vx, road adhesion coefficient and vehicle parameters have been given in the text, so they are not listed in the table. In addition, we have optimized the figures to improve visibility.

Please see the attachment for details.

Thanks again for your comments.Your kind considerations will be greatly appreciated.

Sincerely Yours,

Mr. Xuequan Tang

Round 2

Reviewer 2 Report

Thank you for Authors' answers. I agree with their explanations related to the Subject areas of Electronics Journal. The proposed control method is applied to electrical vehicles. 

I also agree with explanations related to the interest in the proposed subject, even though I did not question that issue. The manuscript shows an interesting issue of cooperation between two electric vehicle motion controllers, this issue is currently a topic that arouses great interest in the science and is important for the development of autonomous vehicles.

In light of these explanations, I believe that the article is suitable for publication in a Electronics Journal.